# A thermostable Cas9 with increased lifetime in human plasma

Lucas B. Harrington[1], David Paez-Espino [2], Brett T. Staahl[1], Janice S. Chen[1], Enbo Ma[1], Nikos C. Kyrpides[2] & Jennifer A. Doudna [1,3,4,5,6]

CRISPR-Cas9 is a powerful technology that has enabled genome editing in a wide range of species. However, the currently developed Cas9 homologs all originate from mesophilic bacteria, making them susceptible to degradation and unsuitable for applications requiring cleavage at elevated temperatures. Here, we show that the Cas9 protein from the thermophilic bacterium *Geobacillus stearothermophilus* (GeoCas9) catalyzes RNA-guided DNA cleavage at elevated temperatures. GeoCas9 is active at temperatures up to 70 °C, compared to 45 °C for *Streptococcus pyogenes* Cas9 (SpyCas9), which expands the temperature range for CRISPR-Cas9 applications. We also found that GeoCas9 is an effective tool for editing mammalian genomes when delivered as a ribonucleoprotein (RNP) complex. Together with an increased lifetime in human plasma, the thermostable GeoCas9 provides the foundation for improved RNP delivery in vivo and expands the temperature range of CRISPR-Cas9.

[1] Department of Molecular and Cell Biology, University of California, Berkeley, CA 94720, USA. [2] Department of Energy, Joint Genome Institute, Walnut Creek, CA 94598, USA. [3] Department of Chemistry, University of California, Berkeley, CA 94720, USA. [4] Howard Hughes Medical Institute, University of California, Berkeley, CA 94720, USA. [5] Innovative Genomics Institute, University of California, Berkeley, CA 94720, USA. [6] MBIB Division, Lawrence Berkeley National Laboratory, Berkeley, CA 94720, USA. Correspondence and requests for materials should be addressed to J.A.D. (email: doudna@berkeley.edu)

The use of CRISPR-Cas9 has rapidly transformed the ability to edit and modulate the genomes of a wide range of organisms[1]. This technology, derived from adaptive immune systems found in thousands of bacterial species, relies on RNA-guided recognition and cleavage of invasive viral and plasmid DNA[2]. The Cas9 proteins from these species differ widely in their size and cleavage activities[3–5]. Despite the abundance and diversity of these systems, the vast majority of applications have employed the first Cas9 homolog developed from *Streptococcus pyogenes* (SpyCas9)[6]. In addition to SpyCas9, several other Cas9 proteins have also been shown to edit mammalian genomes with varying efficiencies[5, 7–10]. While these proteins together provide a robust set of tools, they all originate from mesophilic hosts, making them unsuitable for applications requiring cleavage at higher temperatures or extended protein stability.

This temperature restriction is particularly limiting for genome editing in obligate thermophiles[11]. Recent efforts using SpyCas9 to edit a facultative thermophile have been possible by reducing the temperature within the organism[12]. While effective, this approach is not feasible for obligate thermophiles, and requires additional steps for moderate thermophiles. This is especially important for metabolic engineering for which thermophilic bacteria present enticing hosts for chemical synthesis due to

decreased risk of contamination, continuous recovery of volatile products, and the ability to conduct reactions that are thermodynamically unfavorable in mesophilic hosts[13]. Developing a thermostable Cas9 system will enable facile genome editing in thermophilic organisms using technology that is currently restricted to mesophiles.

CRISPR-Cas9 has also emerged as a potential treatment for genetic diseases[14]. A promising method for the delivery of Cas9 into patients or organisms is the injection of preassembled Cas9 ribonucleoprotein (RNP) complexes into the target tissue or bloodstream[15]. One major challenge to this approach is that Cas9 must be stable enough to survive degradation by proteases and RNases in the blood or target tissue for efficient delivery. Limited protein lifetime will require delivery of higher doses of Cas9 into the patient or result in poor editing efficacy. In contrast, delivering a Cas9 with improved stability could greatly enhance genome-editing efficiency in vivo.

To address these challenges, we tested the thermostable Cas9 protein from *Geobacillus stearothermophilus* (GeoCas9). We find that GeoCas9 maintains activity over a wide temperature range. By harnessing the natural sequence variation of GeoCas9 from closely related species, we engineered a PAM variant that recognizes additional PAM sequences and thereby doubles the number of targets accessible to this system. We also engineered a

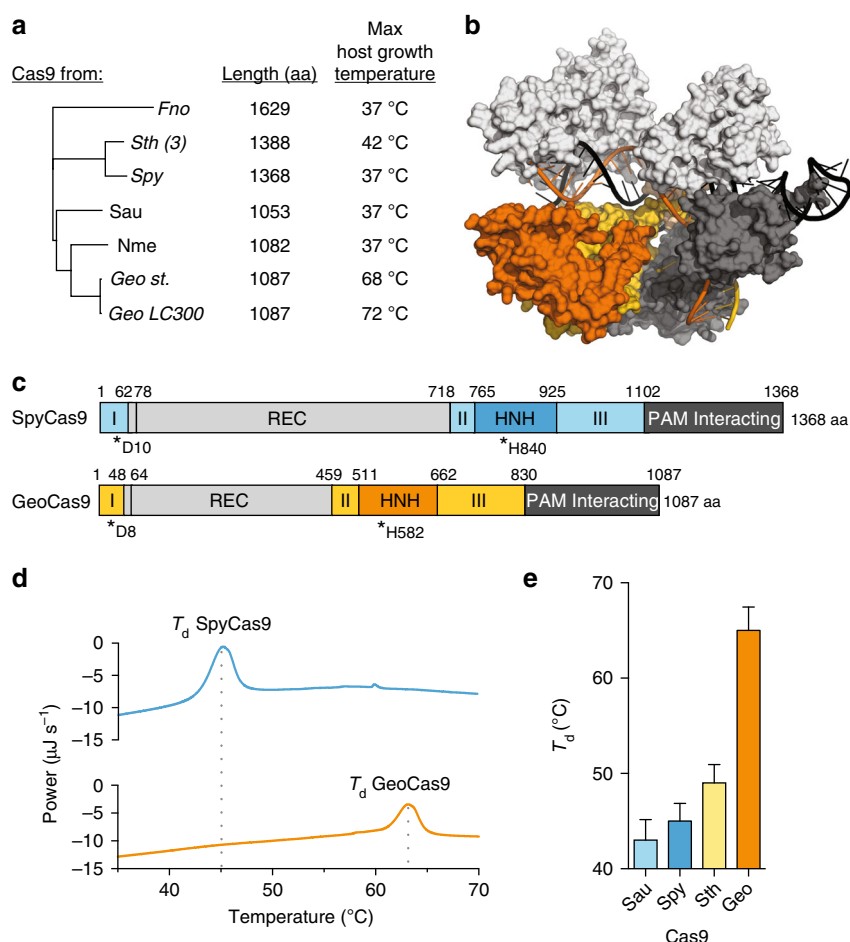

**Fig. 1** GeoCas9 is a thermostable Cas9 homolog. **a** Phylogeny of Cas9 proteins used for genome editing with their length (amino acids) and the maximum temperatures that supports growth of the host indicated to the right[22]. **b** Homology model of GeoCas9 generated using Phyre 2[47] with the DNA from PDB 5CZZ docked in. **c** Schematic illustration of the domains of Spy Cas9 (blue) and GeoCas9 (orange) with active site residues indicated below with asterisks. **d** Representative traces for differential scanning calorimetry (DSC) of GeoCas9 and SpyCas9. **e** Denaturation temperature of various Cas9 proteins as measured by DSC, mean ± S.D. is shown. *Nme Neisseria meningitides, Geo Geobacillus stearothermophilus, Geo LC300 Geobacillus* LC300, *Spy Streptococcus pyogenes, Sau Streptococcus aureus, Fno Francisella novicida, Sth (3) Streptococcus thermophilus* CRISPR III, $T_d$ denaturation temperature

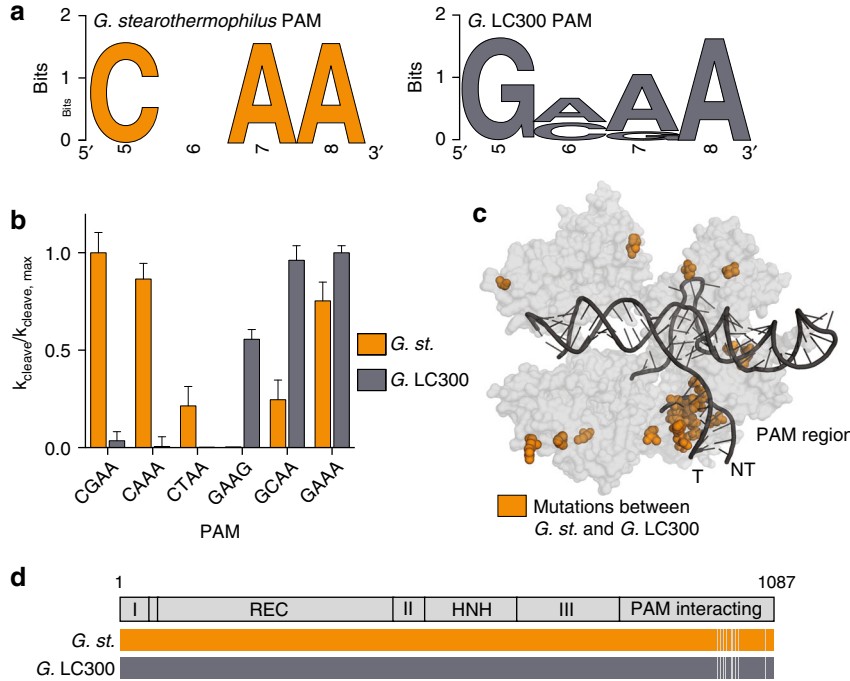

**Fig. 2** PAM identification and engineering of GeoCas9. **a** WebLogo for sequences found at the 3′ end of protospacer targets identified with CRISPRTarget for *Geobacillus stearothermophilus* (left) and *Geobacillus* LC300 (right). **b** Cleavage assays conducted with the two homologs of GeoCas9. Substrates with various PAM sequences were $^{32}$P-labeled and mean ± S.D. is shown. **c** Mapping of mutated residues (orange spheres) between *G. st.* and *G.* LC300 onto the homology model of GeoCas9 showing high density in the PAM-interacting domain near the PAM region of the target DNA. **d** Alignment of the Cas9 proteins from *G. st.* and *G.* LC300 with the domain boundaries shown above. Solid colors represent identical residues and gray lines indicate residues that are mutated between the two Cas9 homologs

highly efficient single-guide RNA (sgRNA) using RNA-seq data from the native organism and show that GeoCas9 can efficiently edit genomic DNA in mammalian cells. The functional temperature range of GeoCas9 complements that of previously developed Cas9 systems, greatly expanding the temperatures that Cas9 can be used for both in vitro cleavage and genome-editing applications.

## Results

**Identification of thermostable Cas9 homologs**. Although thousands of *cas9* homologs have been sequenced, there have been no functionally validated Cas9 from archaea[16], restricting our search for a thermophilic Cas9 to thermophilic bacteria. We searched all the isolates in Integral Microbial Genomes database (IMG) from a thermophilic environment that contained a Cas9-like protein[17] (hits to a TIGRfam model 01,865 for Csn1-like or 03,031 for a Csx12-like). From them, the Cas9 from *G. stearothermophilus* (*G. st.*; formerly *Bacillus stearothermophilus*)[18] stood out because it was full-length and its sequence is shorter than the average Cas9. Most importantly, this candidate is from the organism that can grow in a reported temperature range from 30 to 75 (optimal at 55 °C). A BLASTn of GeoCas9 revealed several nearly identical homologs (from 93.19 to 99.91% identity over the full length) in six other *Geobacillus* species (Supplementary Data 1) and 92.55% identity over the full length in *Effusibacillus pohliae* DSM22757. *G. st.* has been a proven source of enzymes for thermophilic molecular cloning applications[19], thermostable proteases[20], and enzymes for metabolic engineering[21]. Moreover, the wide temperature range that *G. st.* occupies[22] holds promise that the Cas9 from this species (GeoCas9) may be able to maintain activity at both mesophilic and thermophilic temperatures (Fig. 1a). Notably, GeoCas9 is considerably smaller than SpyCas9 (GeoCas9, 1087 amino acids; SpyCas9, 1368 amino

acids). A homology model of GeoCas9 based on available Cas9 crystal structures along with sequence alignments revealed that the small size of GeoCas9 is largely the result of a reduced REC lobe, as is the case with other compact Cas9 homologs from *S. aureus* Cas9 (SauCas9) and *Actinomyces naeslundii* Cas9 (AnaCas9) (Fig. 1b, Supplementary Fig. 1).

We purified GeoCas9 and performed initial thermostability tests using differential scanning calorimetry (DSC), which showed that in the absence of RNA or DNA, GeoCas9 has a denaturation temperature about 20 °C higher than SpyCas9 (Fig. 1d). Moreover, GeoCas9 denatures at 15 °C higher than the slightly thermophilic *S. thermophilus* CRISPR III Cas9 (SthCas9) (Fig. 1e). Given these results, we selected GeoCas9 as a candidate for further development and optimization.

**GeoCas9 PAM identification and engineering**. CRISPR systems have evolved a preference for a protospacer adjacent motif (PAM) to avoid self-targeting of the host genome[23, 24]. These PAM sequences are divergent among Cas9 homologs and DNA targets are often mutated in this region to escape cleavage by Cas9[25]. To identify the PAM for GeoCas9, we first searched for naturally targeted viral and plasmid sequences using CRISPRtarget[26]. The three sequenced strains of *G. st.* provided 77 spacer sequences, and 3 of them had high-confidence viral and plasmid targets (Supplementary Fig. 2, Supplementary Data 2). Extracting the sequences 3′ of the targeted sequence revealed a consensus of 5′-NNNNCNAA-3′ (Fig. 2a, ED Fig. 2). Given the low number of viral targets, we next performed cleavage assays on substrates containing various PAM sequences, revealing a complete PAM sequence of 5′-NNNNCRAA-3′ (Fig. 2b).

In addition to the CRISPR loci found in *G. st.* strains, we also found a type II CRISPR locus in *Geobacillus* LC300 containing a Cas9 with ~97% amino acid identity to the *G. st.* Cas9. Despite

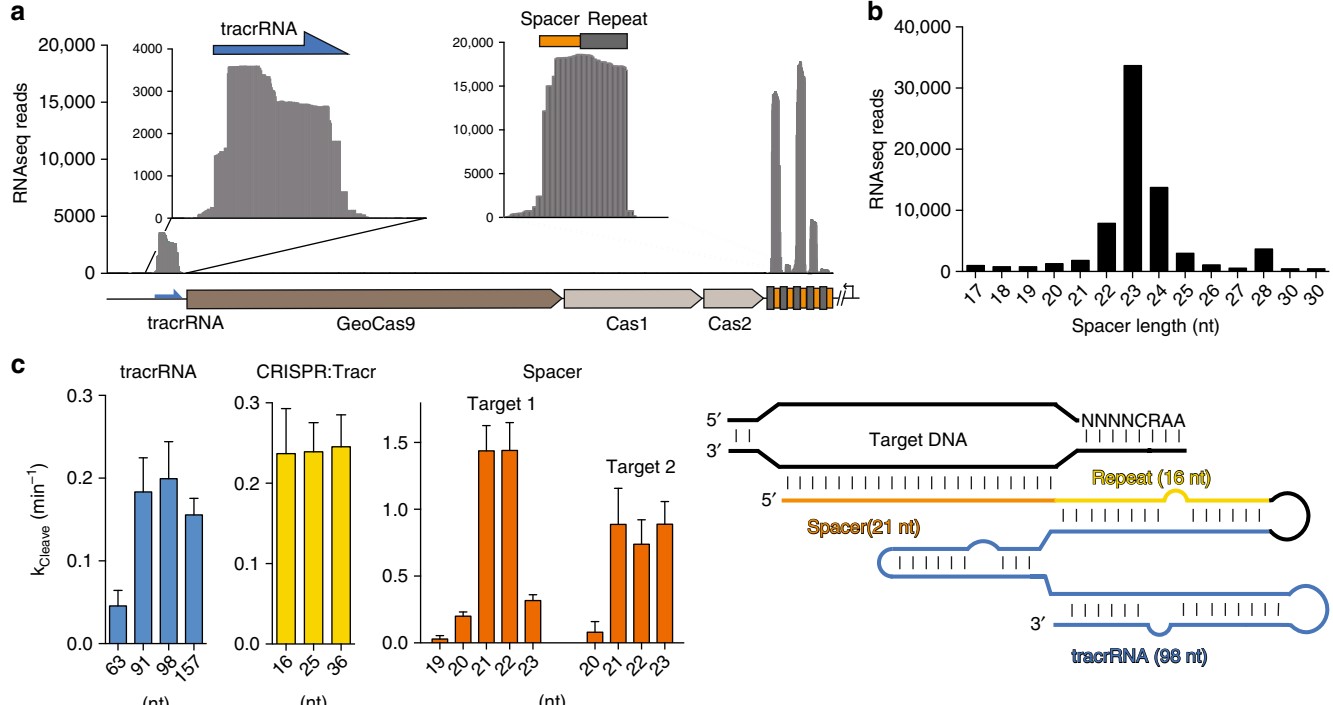

**Fig. 3** Small RNA-seq and sgRNA engineering for GeoCas9. **a** Small RNA sequenced from *G. stearothermophilus* mapped to the CRISPR locus. Inset shows enlargement of the region corresponding to the tracrRNA and the most highly transcribed repeat and spacer sequence. **b** Distribution of the length of the spacer sequences extracted from the small RNA sequencing results. **c** Length optimization of the tracrRNA and crRNA for GeoCas9 and the optimal guide RNA design (right). The length of the tracrRNA, crRNA:tracrRNA duplex and spacer was optimized sequentially by transcribing variations of the sgRNA and testing their ability to guide GeoCas9-mediated cleavage of a radiolabeled substrate. For spacer length, two different targets were used (Target 1 and Target 2). The mean $k_{cleave}$ ± S.D. is shown and experiments were conducted in triplicate

having nearly identical sequences, alignment of these two homologs of GeoCas9 revealed a tight cluster of mutations in the PAM-interacting (PI) domain (Fig. 2d). Furthermore, mapping these mutations onto the homology model of GeoCas9 showed that they are located near the PAM region of the target DNA (Fig. 2c). We hypothesized that this GeoCas9 variant might have evolved altered PAM specificity. By searching for viral targets using the spacers in the *G.* LC300 array, we identified a preference for GMAA in place of the CRAA PAM of *G. st.*, lending support to our hypothesis. We constructed and purified a hybrid Cas9 protein in which the PI domain of the *G.* LC300 Cas9 was substituted for the PI domain of *G. st.* Cas9 and tested cleavage activity on targets containing various PAM sequences (Fig. 2b). We found that, as predicted by protospacer sequences, the hybrid Cas9 preferred a GMAA PAM rather than the CRAA PAM utilized by GeoCas9. Moreover, *G.* LC300 appears to be more specific for its optimal PAM, which may result in lower off-target cleavage for genome-editing applications[27]. By creating a hybrid Cas9 with this naturally occurring PAM-recognition variant, we double the sequence space that can be targeted by GeoCas9 without resorting to structure-based protein engineering as has been done for other Cas9 homologs[27].

**Identification of tracrRNA and engineering of GeoCas9 sgRNA.** CRISPR-Cas9 systems use a trans-activating crRNA (tracrRNA), which is required for maturation of the crRNA and activation of Cas9[6, 28]. To identify the tracrRNA for GeoCas9, we cultured *G. st.* and deep-sequenced the small RNA it produced. We found that the CRISPR array was transcribed despite a lack of phage or plasmid challenge, and that the array was transcribed in the opposite direction of the Cas proteins

(Fig. 3a). The crRNA was processed to 23 nt (Fig. 3b) of the spacer sequence and 18 nt of the repeat sequence in vivo, similar to other small type II-C Cas9 proteins[8, 29]. Mapping of the RNA-seq reads to the CRISPR array also revealed a putative tracrRNA upstream of the Cas9 open reading frame.

We joined this putative tracrRNA to the processed crRNA using a GAAA-tetraloop to generate a sgRNA[6]. Variations of this sgRNA were in vitro transcribed and tested for their ability to direct GeoCas9 to cleave a radiolabeled double-stranded DNA target at 37 °C. We first varied the length of the crRNA:tracrRNA duplex and found that this modification had little impact on the DNA cleavage rate (left, Fig. 3c), making it a valuable place for further sgRNA modifications[30, 31]. Next, we tested the length of the tracrRNA, which here refers to the region after the tetraloop, choosing stopping points near predicted rho-independent terminators. In contrast to the crRNA:tracrRNA duplex length, the length of the tracrRNA had a dramatic effect on the cleavage rate, with sequences shorter than 91 nt supporting only a small amount of cleavage (middle, Fig. 3c). Finally, we varied the length of the spacer sequence and found that 21–22 nt resulted in a more than fivefold increase in cleavage rate, compared to the 20-nt spacer preferred by SpyCas9 (right, Fig. 3c). This finding contrasts with the most abundant spacer length of 23 nt found in RNA-seq. This difference may be due to inter- or intramolecular guide interactions in the in vitro transcribed sgRNA[32]. To test this, we used an additional guide sequence with no predicted structure in the spacer region (Target 2). In contrast to Target 1, Target 2 had similar cleavage rates for 21, 22, and 23 nt (right, Fig. 3c). In addition, testing cleavage of off-target substrates revealed that GeoCas9 has higher sensitivity of mismatched sequences proximal to the PAM than distal, similar to previously described Cas9 proteins (Supplementary Fig. 3a).

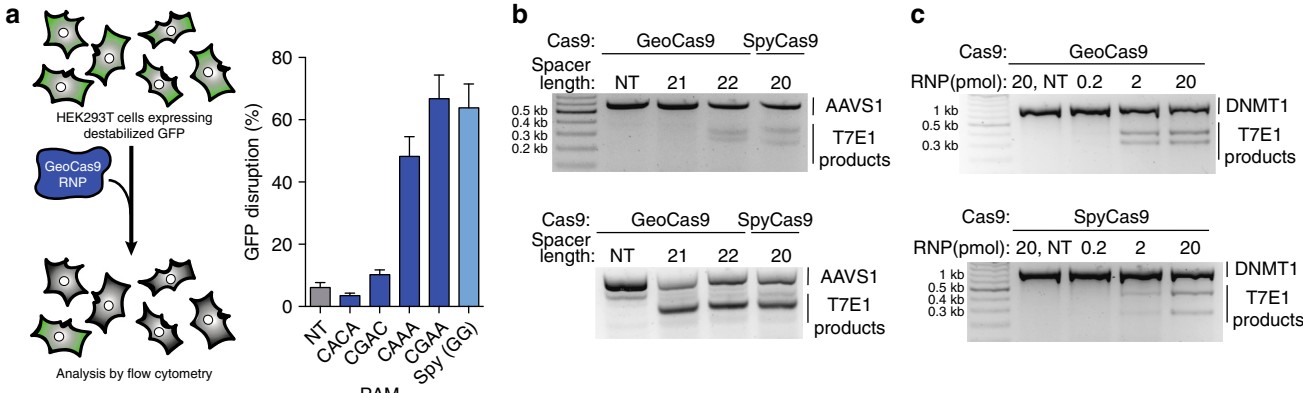

**Fig. 4** Genome-editing activity of GeoCas9 in mammalian cells. **a** EGFP disruption in HEK293T cells by GeoCas9. HEK293FT cells expressing a destabilized GFP were transfected with GeoCas9 RNP preassembled with a targeting or non-targeting guide RNA. Cells were analyzed by flow cytometry and targets adjacent to the CRAA PAM resulted in efficient GFP disruption. **b** T7E1 analysis of indels produced at the AAVS1 locus when the guide length was varied from 21 to 22 nt. The Cas9 used is indicated above each lane and the length of the spacer is shown below. **c** T7E1 analysis of indels produced using a titration of GeoCas9 and SpyCas9 RNP targeting the DNMT1 locus in HEK293T cells. The Cas9 used is indicated above each lane and the amount of RNP delivered to each well of a 96-well plate is indicated below. *NT* non-targeting

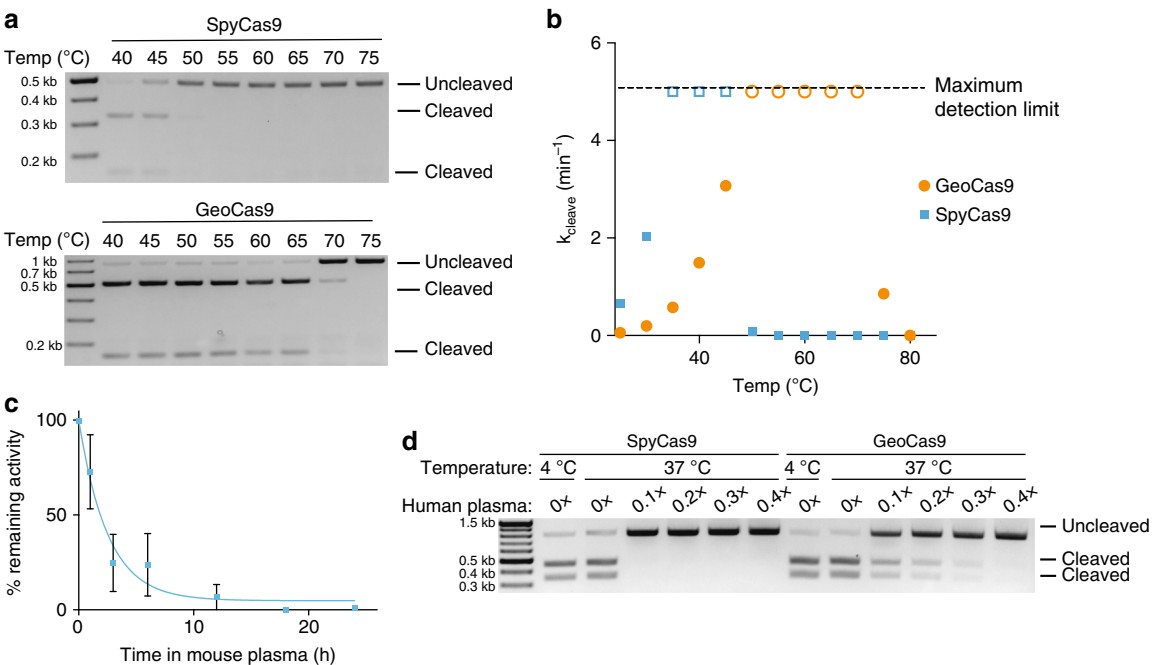

**Fig. 5** Thermostability of GeoCas9 and longevity in human plasma. **a** Activity of SpyCas9 and GeoCas9 after incubation at the indicated temperature. After challenging at the higher temperature, reactions were conducted at 37 °C using a 1:1 ratio of substrate to RNP. **b** Cleavage rate of SpyCas9 and GeoCas9 RNPs at various temperatures. Maximum detection limit is shown by the dashed line at $k_{cleave} = 5$, indicating that the reaction completed in ≤30 s. **c** Lifetime of SpyCas9 after incubation in 50% mouse plasma. Incubation was done at 37 °C for the specified amount of time after which DNA substrate was introduced at an equimolar ratio of substrate to RNP. **d** Effect of incubating GeoCas9 and SpyCas9 in human plasma. After incubation in varying concentrations of human plasma for 8 h at 37 °C, the reaction was carried out with 1:1 ratio of DNA substrate to RNP

**Genome editing by GeoCas9 RNP in mammalian cells**. With evidence that GeoCas9 maintains cleavage activity at mesophilic temperatures, we assessed the ability of GeoCas9 to edit mammalian genomes. We tested GeoCas9 and SpyCas9 editing efficiency by delivering preassembled RNP complexes into cultured cells, circumventing differences between SpyCas9 and GeoCas9 protein expression. First, GeoCas9 RNPs targeting regions adjacent to various PAM sequences were delivered into HEK293T cells expressing a destabilized GFP (Fig. 4a). We found that when targeted to sequences adjacent to the preferred CRAA

PAM, GeoCas9 decreased GFP fluorescence at levels comparable to those observed for SpyCas9 (Fig. 4a). Next, we targeted Geo-Cas9 to cleave the native genomic loci DNMT1 and AAVS1 (Fig. 4b, c). We varied the length of the targeting spacer sequence and found that at one site 21 nt was a sufficient length to efficiently induce indels, while at another site a 22-nt spacer length was necessary. Given this variability and that extending the spacer length to 22 nt had no detrimental effects, we conclude that a 22-nt guide segment length is preferred for use in genome-editing applications. Moreover, when we tested editing efficiency at a site

containing an overlapping PAM for both GeoCas9 and SpyCas9, we observed similar editing efficiencies by both proteins (Fig. 4b). At the DNMT1 locus, we titrated amounts of GeoCas9 and SpyCas9 RNPs to assess the effect on genome editing efficiency (Fig. 4c). Products analyzed by T7E1 assay again showed efficient production of indels by both GeoCas9 and SpyCas9. These results demonstrate that GeoCas9 is an effective alternative to SpyCas9 for genome editing in mammalian cells.

**Stability of GeoCas9.** Based on initial observations showing that GeoCas9 protein remains folded at elevated temperatures (Fig. 1d, e), we tested whether the GeoCas9 RNP maintains activity after exposure to high temperatures. We incubated Spy-Cas9 and GeoCas9 at a challenge temperature and added equimolar substrate to test the fraction of RNP that remained functional. After incubation for 10 min at 45 °C, the fraction of active SpyCas9 was greatly reduced (Fig. 5a). In contrast, the fraction of GeoCas9 after incubation at 45 °C remained at 100% and not until challenge at 70 °C did we detect a decrease in activity (Fig. 5a).

Often thermostability comes at the cost of reduced activity at lower temperatures[33]. However, the wide range of natural growth temperatures for *G. st.* suggested that GeoCas9 might maintain activity at mesophilic temperatures. To examine this hypothesis, we measured the cleavage rate of SpyCas9 and GeoCas9 at various temperatures (Fig. 5b). SpyCas9 DNA cleavage rates increased between 20 and 35 °C, reaching maximum levels from 35 to 45 °C. Above these temperatures, SpyCas9 activity dropped sharply to undetectable levels, as predicted by thermostability measurements. In contrast, GeoCas9 activity increased to the maximum detection limit at 50 °C and maintained maximum detectable activity up to 70 °C, dropping to low levels at 75 °C. These results make GeoCas9 a valuable candidate for editing obligate thermophilic organisms and for biochemical cleavage applications requiring Cas9 to operate at elevated temperatures.

The lifetime of proteins in blood is often limiting for their use as a therapeutic and many strategies have been employed to improve this, including fusion of other proteins and polymers to the protein of interest[34, 35]. We investigated the lifetime of SpyCas9 RNP in mouse plasma and found that it had a half-life of ~1.5 h (Fig. 5c). Further investigation revealed that a combination of RNA and protein degradation likely plays a role in this inactivation (Supplementary Fig. 3b, c). Although what determines the lifetime of a protein in blood is complex, it was shown previously that thermostabilization of a protein can increase its lifetime in blood[36]. To test if this is the case for GeoCas9, we incubated SpyCas9 and GeoCas9 in diluted human plasma at 37 °C for 8 h and measured the amount of Cas9 activity remaining (Fig. 5d). Although SpyCas9 maintained activity when incubated in reaction buffer at 37 °C, its activity was abolished even at the lowest concentration of plasma. In contrast, GeoCas9 maintained significant activity after incubation with human plasma, making it a promising candidate for in vivo RNP delivery.

## Discussion

Our results establish GeoCas9 as a thermostable Cas9 homolog and expand the temperatures at which Cas9 can be used. We anticipate that the development of GeoCas9 will enhance the utility of CRISPR-Cas9 technology at both mesophilic and thermophilic temperatures. The ability of Cas9 to function reliably in a wide range of species has been key to its rapid adoption as a technology, but the previously developed Cas9 homologs are limited for use in organisms that can grow below 42 °C. The complementary temperature range of GeoCas9 with SpyCas9 (Fig. 5b) opens up Cas9-based genome editing to obligate

thermophiles and facultative thermophiles, without the additional steps of altering the temperature of the organisms. We also anticipate that GeoCas9 will be useful for in vitro molecular biology applications requiring targeted cleavage at elevated temperatures. Furthermore, we predict that the extended lifetime of GeoCas9 in human plasma may enable more efficient delivery of Cas9 RNPs. GeoCas9 also has an extended PAM and spacer sequence compared to SpyCas9. These features are similar to SauCas9 and NmeCas9, both of which have been shown to have naturally higher fidelity than SpyCas9[5, 37–39] and it will be interesting for future work to investigate if this trend also holds true for GeoCas9.

We were interested to note that GeoCas9 and SpyCas9 induced similar levels of indels in HEK293T cells as SpyCas9 when delivered as an RNP (Fig. 4b–d), despite GeoCas9's lower DNA cleavage rate at 37 °C (Fig. 5b). We conclude that biochemical cleavage rates may not reflect the limiting step of target search in a human cell. It may be that GeoCas9 can persist longer in cells, which raises its effective concentration over time and compensates for its slower cleavage rate. Moreover, in applications requiring delivery of Cas9 into the bloodstream, the benefit of improved stability by GeoCas9 may become even more apparent. However, it remains to be seen if the efficient GeoCas9-mediated genome editing observed here will hold true in more challenging genome-editing applications.

The development of GeoCas9 hinged upon utilizing the naturally occurring diversity of CRISPR systems. The sheer abundance and diversity of Cas9 makes it advantageous over newer type V systems, such as Cpf1, for developing specialized genome editing tools. It has previously been suggested that type II CRISPR systems are only found in mesophilic bacteria, and that protein engineering would be required to develop a thermophilic Cas9[40]. The rarity of type II CRISPR systems in thermophiles is surprising given that CRISPR systems in general are enriched in thermophilic bacteria and archaea[41]. However, by searching the continually growing number of sequenced bacteria, we uncovered a naturally occurring thermophilic Cas9. Exploiting Cas9 sequence diversity, rather than engineering thermostability, revealed a protein that maintains activity over a broad temperature range, which is often difficult to select for using directed evolution. Using the natural context of this CRISPR system, including the transcribed RNA and targeted sequences, we further developed GeoCas9 with minimal experimental optimization. The strategy of mining the natural context and diversity of CRISPR systems has proven successful for uncovering novel interference proteins[16, 42], and we anticipate that it can be applied more broadly to discover and develop new genome editing tools.

## Methods

**Identification of Cas9 homologs and generation of plasmids.** We mined all isolate genomes from the public IMG database[17] using the "Genome Search by Metadata Category tool." We selected all the genomes annotated as "thermophile" (336) or "hyperthermophile" (94) and searched for the presence of Cas9-like candidates (hits to a TIGRfam model 01,865 for Csn-like or 03,031 for a Csx12-like) contained within a full CRISPR-Cas system (presence of Cas1, Cas2, and a repeat-spacers array). We initially selected the GeoCas9 variant due to its completeness, smaller gene size (shorter than the widely used SpyCas9), and growth in a large temperature range from 30 to 75 (optimal at 55 °C). The Cas9 from *G. stearothermophilus* was codon optimized for *E. coli*, ordered as Gblocks from Integrated DNA Technologies (IDT), and assembled using Gibson Assembly. For protein expression, a pET-based plasmid containing an N-terminal 10xHis-tag and Maltose Binding Protein (MBP) was used. For PAM depletion assays, a p15A plasmid was generated with the sgRNA constitutively expressed. The plasmids used are available from Addgene and their maps can be found in Supplementary Data 3.

**Cas9 purification.** Cas9 was purified as previously described[6] with modification. After induction, *E. coli* BL21(DE3) expressing Cas9 was grown in terrific broth overnight at 18 °C. Cells were collected, re-suspended in lysis buffer (50 mM Tris-HCl, pH 7.5, 20 mM imidazole, 0.5 mM TCEP, 500 mM NaCl, 1 mM PMSF),

broken by sonication, and purified on Ni-NTA resin. TEV was added to the elution and allowed to cleave overnight at 4 °C. The resulting protein was loaded over tandem columns of an MBP affinity column onto a heparin column and eluted with a linear gradient from 300 to 1250 mM NaCl. The resulting fractions containing Cas9 were purified by gel filtration chromatography and flash frozen in storage buffer (20 mM HEPES-NaOH pH 7.5, 5% glycerol, 150 mM NaCl, 1 mM TCEP).

**Differential scanning calorimetry**. Cas9 proteins were dialyzed into degassed DSC buffer (0.5 mM TCEP, 50 mM $KH_2PO_4$ (pH 7.5), 150 mM NaCl) overnight at 4 °C. Samples were diluted to 0.3 mg ml$^{-1}$ and loaded a sample cell of a NanoDSC (TA instruments); buffer alone was used in the reference cell. The cell was pressurized to 3 atm and the sample was heated from 20 to 90 °C. Measurements made for buffer in both the sample and reference cells were subtracted from the sample measurements.

**Biochemical cleavage assays**. Radioactive cleavage assays were conducted as previously described[43]. Briefly, reactions were carried out in 1 × reaction buffer (20 mM Tris-HCl, pH 7.5, 100 mM KCl, 5 mM $MgCl_2$, 1 mM DTT, and 5% glycerol (v v$^{-1}$)). Cas9 of 100 nM and 125 nM sgRNA were allowed to complex for 5 min at 37 °C. Approximately 1 nM radiolabeled probe was added to the RNP to initiate the reaction. Finally, the reaction was quenched with 2 × loading buffer (90% formamide, 20 mM EDTA, 0.02% bromophenol blue, 0.02% xylene cyanol, and products were analyzed on 10% urea-PAGE gel containing 7 M urea).

For thermostability measurements (Fig. 4a), 100 nM Cas9 was complexed with 150 nM sgRNA in 1 × reaction buffer for 5 min at 37 °C. PCR product of 100 nM containing the targeted sequence was cleaved using dilutions of the estimated 100 nM RNP complex to accurately determine a 1:1 ratio of Cas9 to target. Next, samples were challenged at the indicated temperature (40–75 °C) for 10 min and then returned to 37 °C. PCR product of 100 nM containing the targeted sequence was added to the reaction and it was allowed to react for 30 min at 37 °C. The reaction was quenched with 6 × quench buffer (15% glycerol (v v$^{-1}$), 1 mg ml$^{-1}$ Orange G, 100 mM EDTA) and products were analyzed on a 1.25% agarose gel stained with ethidium bromide.

For thermophilicity measurements (Fig. 4b), 500 nM Cas9 was complexed with 750 nM sgRNA in 1× reaction buffer for 5 min at 37 °C. The samples were placed at the assayed temperature (20–80 °C) and 100 nM of PCR product was added to the reaction. Time points were quenched using 6 × quench buffer and analyzed on a 1.25% agarose gel stained with SYBR Safe (Thermo Fisher Scientific).

To study the effect of human plasma on the stability of Cas9 proteins, preassembled Cas9-RNP was incubated for 8 h either at 37 °C or 4 °C in reaction buffer with the specified amount of normal human plasma. Substrate was then added and cleavage products were analyzed as described for thermostability measurements. RNA sequences and DNA targets can be found in Supplementary Data 3. Uncropped gels related to Figs. 4 and 5 can be found in Supplementary Fig. 4.

**Small RNA sequencing**. G. stearothermophilus was obtained from ATCC and cultured at 55 °C in nutrient broth (3 g beef extract and 5 g peptone per liter water, pH 6.8) to saturation. Cells were pelleted and RNA was extracted using hot phenol extraction as previously described[44]. Total RNA was treated with TURBO DNase (Thermo Fisher Scientific), rSAP (NEB), and T4 PNK (NEB) according to manufactures instructions. Adapters were ligated onto the 3′ and 5′ ends of the small RNA, followed by reverse transcription with Superscript III. The library was amplified with limited cycles of PCR, gel-extracted on an 8% native PAGE gel, and sequenced on an Illumina MiSeq. Adapters were trimmed using Cutadapt and sequences > 10nt were mapped to the G. st. CRISPR locus using Bowtie 2[45].

**HEK293T EGFP disruption assay and indel analysis**. HEK293T cells expressing a destabilized GFP were grown in Dulbecco's Modification of Eagle's Medium (DMEM) with 4.5 g l$^{-1}$ glucose L-glutamine and sodium pyruvate (Corning Cellgro), supplemented with 10% fetal bovine serum, penicillin, and streptomycin at 37 °C with 5% $CO_2$. Approximately 24 h before transfection, ~3 × 10$^4$ cells were seeded into each well of a 96-well plate. The next day, 20 pmol (unless otherwise specified) of RNP was assembled as previously described[46] and mixed with 10 μl OMEM. The RNP was added to 10 μl of 1:10 dilution of Lipofectamine 2000 (Life Technologies) in OMEM and allowed to incubate at room temperature for 10 min and added to the cells. Cells were analyzed for GFP fluorescence 48 h later using Guava EasyCyte 6HT. Experiments were conducted in triplicate and the mean ± S. D. is shown. For analysis of indels, genomic DNA was extracted using Quick Extraction Solution (Epicentre), and the DNMT1 and AAVS1 loci were amplified by PCR. T7E1 reaction was conducted according to the manufacturer's instructions and products were analyzed on a 1.5% agarose gel stained with SYBR gold (Thermo Fisher Scientific).

**Data availability**. Plasmids used in this study are available on Addgene (#87700, #87703). RNA sequencing data is available on the NCBI Sequence Read Archive under accession code PRJNA413627.

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

## Acknowledgements

We thank A. Tambe, N. Ma, and K. Zhou for technical assistance and discussions. L.B.H. and J.S.C. are supported by US National Science Foundation Graduate Research Fellowships. J.A.D. is an investigator of the Howard Hughes Medical Institute. This research was supported in part by the Allen Distinguished Investigator Program, through The Paul G. Allen Frontiers Group and the National Science Foundation (MCB-1244557 to J. A.D.).

## Author contributions

L.B.H. and B.T.S. designed and executed experiments with help from J.S.C., E.M. and J.A.D.; the search for thermophilic Cas9 homologs was conducted by D.P.-E. and N.C.K. All authors revised and agreed to the manuscript.
