## [Peer Review File · Nature Communications]

Reviewers' Comments:

Reviewer #1:

Remarks to the Author:

Harrington et al. characterize a thermostable Cas9 protein from *Geobacillus stearothermophilus* as a new genome editing tool. This work demonstrates GeoCas9 is active at a range of temperatures up to 70°C which will facilitate many possible applications, is active in human cells and intriguingly also show that GeoCas9 is active in the presence of human serum while SpyCas9 is not. Importantly, the authors present GeoCas9 as a ready to use tool in that they have characterized two PAM variants, identified the tracrRNA, demonstrated a fused sgRNA will work with GeoCas9 and determined a semi-optimal protospacer length. This paper is excellently executed and will be broadly interesting to the genome engineering community as a new robust gene editing option.

Major consideration:

The observation that SpyCas9 is unstable in serum while GeoCas9 is stable in serum is underexplored in this paper. To my knowledge this observation has not been previously published. Is inactivation due to proteolysis of SpyCas9 or is there an endogenous protein that binds to and inhibits SpyCas9 in some way? It would be fairly straightforward to check whether SpyCas9 is cleaved by a protease by western blot. If SpyCas9 is not inactivated through proteolysis it would add to the impact of the paper if the mechanism of inactivation was explored. This point has wide ranging implications and will be widely cited. Does this observation also hold true in the serum of other mammals such as mice or cows? Fetal bovine serum is commonly used in tissue culture media and some buffers. Similarly it would greatly add to the impact of this paper if similar characterization of proteolysis could be done as a control for GeoCas9 in serum.

Minor concerns:

Please consider citing PMID:23907171 in addition to ref 31 as this the first paper to my knowledge to modify the sgRNA to encode additional functions beyond genome editing or binding.

The observation that a 23bp protospacer is 5 fold less active than a 21-22bp protospacer is quite interesting. If the increase in activity in 3C is due to inter- or intramolecular interactions then this should be protospacer sequence dependent. The authors should remove speculation or test 2-3 additional protospacers with different protospacer sequences of 23 or 21 bp length to support this claim. Or the authors could use an RNA structure prediction algorithm to support this claim.

Any demonstration of mismatch tolerance between the sgRNA and target DNA sequence would preliminarily characterize off target activity and suggest rules for avoiding off target activity.

The authors comment that thermostable proteins have a longer half-life in the blood but this is perhaps oversimplified as long flexible peptides can also extend the half life of proteins and peptides in vivo (PMID:19915550).

Reviewer #2:

Remarks to the Author:

This is a very interesting manuscript from Doudna and colleagues that describes a thermostable Cas9 protein. Until now, the available CRISPR-Cas9 tools all function within a similar range of temperatures, which restricts their applications. Harrington et al tested two homologs of Cas9 from the thermophilic bacterium *Geobacillus* in order to identify a more thermostable Cas9 variant. They characterized the activity of GeoCas9, including defining the PAM sequence for each, making a hybrid GeoCas9 to recognize a second PAM, and defining the optimal length of each component of the guide RNA. They show that GeoCas9 is able to cleave targets in eukaryotic host cells, that its optimal range of activity is at ~15-20°C higher temperatures than SpyCas9, and that it has

improved stability in human plasma. A thermostable Cas9 expands the Cas9 toolbox, with a compelling application being the potential to engineer thermophilic bacteria that grow in extreme environments.

Stability in human plasma is presented as the main selling point for GeoCas9. While there is some GeoCas9 activity after 8 hours in 10-20% human plasma vs none from SpyCas9, there is little evidence that this would have extended cleavage activity in the blood, which is what it sounds like this assay is the proxy for. What does the timecourse of cleavage activity look like after incubation in vitro in human blood, rather than the mixture of cleavage buffer and plasma?

It would also be helpful to have more discussion of whether the much slower cleavage rate of GeoCas9 at 37C than SpyCas9, affected time-to-cleavage in host cells and following incubation in the plasma mixture. It was hard to understand the reasons for the strong cleavage activity results for GeoCas9 at 37C in figure 4 vs the relatively slow cleavage rate of GeoCas9 at 37C in Figure 5b.

Minor points:

-Is the "=" a typo on line 88 of page 4?

-In the legend for Figure 5, "b" is not bolded

-Is "robust" an appropriate descriptor for GeoCas9 activity after incubation in 0.1x plasma?

-Fig 5A – why wasn't 37C included in the stability assay as a control for a physiologically relevant condition and a more comprehensive comparison to SpyCas9; why did the temperatures start at 40?

-In Figures 4B and C, it is unclear from the legend and methods what we are looking at in the top panel or each cleavage assay.

-In the legend of Figure 3C, it was unclear whether tracrRNA length is referring to the optimal length of the entire guideRNA, the tracrRNA tail after the spacer, or the entire tracrRNA length following the linkage to the crRNA.

Reviewers' comments:

Reviewer #1 (Remarks to the Author):

Harrington et al. characterize a thermostable Cas9 protein from *Geobacillus stearothermophilus* as a new genome editing tool. This work demonstrates GeoCas9 is active at a range of temperatures up to 70C which will facilitate many possible applications, is active in human cells and intriguingly also show that GeoCas9 is active in the presence of human serum while SpyCas9 is not. Importantly, the authors present GeoCas9 as a ready to use tool in that they have characterized two PAM variants, identified the tracrRNA, demonstrated a fused sgRNA will work with GeoCas9 and determined a semi-optimal protospacer length. This paper is excellently executed and will be broadly interesting to the genome engineering community as a new robust gene editing option.

Major consideration:

The observation that SpyCas9 is unstable in serum while GeoCas9 is stable in serum is underexplored in this paper. To my knowledge this observation has not been previously published. Is inactivation due to proteolysis of SpyCas9 or is there an endogenous protein that binds to and inhibits SpyCas9 in some way? It would be fairly straightforward to check whether SpyCas9 is cleaved by a protease by western blot. If SpyCas9 is not inactivated through proteolysis it would add to the impact of the paper if the mechanism of inactivation was explored. This point has wide ranging implications and will be widely cited. Does this observation also hold true in the serum of other mammals such as mice or cows? Fetal bovine serum is commonly used in tissue culture media and some buffers. Similarly it would greatly add to the impact of this paper if similar characterization of proteolysis could be done as a control for GeoCas9 in serum.

Response: We agree that much work remains to be done regarding the lifetime of SpyCas9 in serum. In response to your suggestion we tested the stability of the guide RNA and SpyCas9 protein in plasma. We find that in the context of a Cas9 RNP, the guide RNA is largely degraded after incubation in human plasma. Western blotting shows that SpyCas9 protein is degraded slightly after incubation in human plasma. Inactivation of Cas9 in blood is likely due to a combination of factors including proteolysis, RNase cleavage and immune response. While interesting, we feel that this is best studied in an *in vivo* context and is beyond the scope of this work.

Minor concerns:

Please consider citing PMID:23907171 in addition to ref 31 as this the first paper to my knowledge to modify the sgRNA to encode additional functions beyond genome editing or binding.

Response: This reference has been added.

The observation that a 23bp protospacer is 5 fold less active than a 21-22bp protospacer is quite interesting. If the increase in activity in 3C is due to inter- or intramolecular interactions then this should be protospacer sequence dependent. The authors should remove speculation or test 2-3 additional protospacers with different protospacer sequences of 23 or 21 bp length to support this claim. Or the authors could use an RNA structure prediction algorithm to support this claim.

Response: As suggested, we tested an additional guide sequence with no predicted structure of the spacer sequence and find that a 23-nt guide segment works as efficiently as a 21-nt or 22-nt guide segment. We have modified the text and figure to represent this.

Any demonstration of mismatch tolerance between the sgRNA and target DNA sequence would preliminarily characterize off target activity and suggest rules for avoiding off target activity.

Response: We tested mismatches in the seed region and at the PAM distal end of the guide sequence and find that GeoCas9 seems to have a similar seed region to those of previously reported Cas9 orthologs. While it is likely that the longer PAM and spacer sequence of GeoCas9 will result in lower off-targets as reported for NmeCas9 and SauCas9, this will require further investigation. We have added references to direct readers to off-target studies for these other Cas9 systems with longer PAM and spacer sequences.

The authors comment that thermostable proteins have a longer half-life in the blood but this is perhaps oversimplified as long flexible peptides can also extend the half-life of proteins and peptides *in vivo* (PMID:19915550).

Response: We agree that determinants of protein lifetime in blood are complex. The cited reference showed that for the same protein, thermostabilization leads to increased lifetime in the blood. We have clarified the text and added a few more references to acknowledge this complexity.

Reviewer #2 (Remarks to the Author):

This is a very interesting manuscript from Doudna and colleagues that describes a thermostable Cas9 protein. Until now, the available CRISPR-Cas9 tools all function within a similar range of temperatures, which restricts their applications. Harrington et al tested two homologs of Cas9 from the thermophilic bacterium *Geobacillus* in order to identify a more thermostable Cas9 variant. They characterized the activity of GeoCas9, including defining the PAM sequence for each, making a hybrid GeoCas9 to recognize a second PAM, and defining the optimal length of each component of the guide RNA. They show that GeoCas9 is able to cleave targets in eukaryotic host cells, that its optimal range of activity is at ~15-20°C higher temperatures than SpyCas9, and that it has improved stability in human plasma. A thermostable Cas9 expands the Cas9 toolbox, with a compelling application being the potential to engineer thermophilic bacteria that grow in extreme environments.

Stability in human plasma is presented as the main selling point for GeoCas9. While there is some GeoCas9 activity after 8 hours in 10-20% human plasma vs none from SpyCas9, there is little evidence that this would have extended cleavage activity in the blood, which is what it sounds like this assay is the proxy for. What does the timecourse of cleavage activity look like after incubation *in vitro* in human blood, rather than the mixture of cleavage buffer and plasma?

Response: We originally tested GeoCas9 stability in diluted human plasma because this solution was more amenable to analysis on an agarose gel. Tissue culture typically employs diluted serum and we think this is an important consideration for passive uptake of Cas9 (as in reference 15). As suggested, we conducted a time-course of SpyCas9 incubation in mammalian plasma and further investigated the degradation of the protein and RNA in the context of a Cas9 RNP. Inactivation of Cas9 in blood is likely due to a combination of factors including proteolysis, RNase cleavage and immune response. While interesting, we feel that this is best studied in an *in vivo* context and is beyond the scope of this work.

It would also be helpful to have more discussion of whether the much slower cleavage rate of GeoCas9 at 37°C than SpyCas9, affected time-to-cleavage in host cells and following incubation in the plasma mixture. It was hard to understand the reasons for the strong cleavage activity results for GeoCas9 at 37°C in figure 4 vs the relatively slow cleavage rate of GeoCas9 at 37°C in Figure 5b.

Response: We agree that this result is surprising, and think it highlights that cleavage of a short DNA substrate, which is what is tested in Figure 5b, is not limiting in most genome editing applications. Instead, the search process of investigating each accessible PAM in a genome is likely the rate-limiting step. It is also possible that the extended lifetime of GeoCas9 compensates for its slower cleavage rate. We also acknowledge that editing HEK293T is a good starting point for testing a new genome editing tool, but it remains to be seen if this efficient genome editing will hold true for more challenging genome editing applications, and we have modified the text to reflect this.

Minor points:

-Is the "=" a typo on line 88 of page 4?

Response: Thank you for identifying this mistake, it has been removed.

-In the legend for Figure 5, "b" is not bolded

Response: Thank you for identifying this mistake, it has now been corrected.

-Is “robust” an appropriate descriptor for GeoCas9 activity after incubation in 0.1x plasma?

Response: We have removed this word in our description of GeoCas9 activity after incubation in plasma.

-Fig 5A – why wasn't 37C included in the stability assay as a control for a physiologically relevant condition and a more comprehensive comparison to SpyCas9; why did the temperatures start at 40?

Response: Most of the other assays are done at 37C and so we thought this information was redundant with other figures. For example, the controls for Figure 5d (previously 5c) show that both GeoCas9 and SpyCas9 are active after incubation at 37C in buffer for 8 hrs.

-In Figures 4B and C, it is unclear from the legend and methods what we are looking at in the top panel or each cleavage assay.

Response: We have expanded the figure legend to clarify what the figure is showing.

-In the legend of Figure 3C, it was unclear whether tracrRNA length is referring to the optimal length of the entire guideRNA, the tracrRNA tail after the spacer, or the entire tracrRNA length following the linkage to the crRNA.

Response: The sequence for the various guideRNA sequences tested is available in the supplemental table. Here the tracrRNA is referring to the sequence in the sgRNA after the tetraloop. We have added a clarification to the text to make this clear.

Reviewers' Comments:

Reviewer #1:

Remarks to the Author:

The authors have satisfied all my concerns and I believe the paper should be published as is. This new gene editor system will be widely useful.

Reviewer #2:

Remarks to the Author:

The authors have addressed all the queries. I have no further comments.